# Epidemiological transition to mortality and refracture following an initial fracture

**Thao Phuong Ho-Le**[1,2,3]\*, **Thach S Tran**[1,4], **Dana Bliuc**[1,4], **Hanh M Pham**[1,5], **Steven A Frost**[1], **Jacqueline R Center**[1,4], **John A Eisman**[1,4,6], **Tuan V Nguyen**[1,4,6,7]\*

[1]Healthy Ageing Theme, Garvan Institute of Medical Research, Darlinghurst, Australia; [2]Swinburne University of Technology, Melbourne, Australia; [3]Faculty of Engineering and Information Technology, Hatinh University, Hatinh, Viet Nam; [4]St Vincent Clinical School, UNSW Sydney, Sydney, Australia; [5]Fertility Department, Andrology and Fertility Hospital of Hanoi, Hanoi, Viet Nam; [6]School of Medicine Sydney, University of Notre Dame Australia, Sydney, Australia; [7]School of Biomedical Engineering, University of Technology, Sydney, Australia

**Abstract** This study sought to redefine the concept of fracture risk that includes refracture and mortality, and to transform the risk into "skeletal age". We analysed data obtained from 3521 women and men aged 60 years and older, whose fracture incidence, mortality, and bone mineral density (BMD) have been monitored since 1989. During the 20-year follow-up period, among 632 women and 184 men with a first incident fracture, the risk of sustaining a second fracture was higher in women (36%) than in men (22%), but mortality risk was higher in men (41%) than in women (25%). The increased risk of mortality was not only present with an initial fracture, but was accelerated with refractures. Key predictors of post-fracture mortality were male gender (hazard ratio [HR] 2.4; 95% CI, 1.79–3.21), advancing age (HR 1.67; 1.53–1.83), and lower femoral neck BMD (HR 1.16; 1.01–1.33). A 70-year-old man with a fracture is predicted to have a skeletal age of 75. These results were incorporated into a prediction model to aid patient-doctor discussion about fracture vulnerability and treatment decisions.

\*For correspondence:
t.ho-le@garvan.org.au (TPH-L);
t.nguyen@garvan.org.au (TVN)

## Introduction

Fracture due to osteoporosis imposes a significant health care burden to the society. From the age of 50, the residual lifetime risk of fracture is ~50% in women and ~30% in men (***Nguyen et al., 2007a***). In women, the lifetime risk of hip fracture is actually equivalent to or higher than the risk of invasive breast cancer (***Nguyen et al., 2007a***; ***Cummings et al., 1989***). In men, the lifetime risk of hip and vertebral fractures (17%) is comparable to the lifetime risk of being diagnosed with prostate cancer (***Cummings et al., 1989***; ***Shortt and Robinson, 2005***). In the United States alone, the cost attributable to osteoporosis and fracture was estimated to be $22 billion (2008), higher than the cost attributable to breast cancer (***Blume and Curtis, 2011***). With the rapid aging of the population, the burden of osteoporosis and fracture will become much more pronounced worldwide.

Various studies have found that an existing fracture signals an increased risk of subsequent fracture and/or mortality (***Shortt and Robinson, 2005***; ***Center et al., 2007***; ***Bliuc et al., 2009***). However, the sequential consequences of fracture, recurrent fracture, and mortality are highly heterogeneous among individuals, and it is not clear why some individuals do well after an initial fracture, but others go on to sustain a refracture and mortality. Existing fracture risk assessment tools focus on predicting the risk of an initial fracture (***Kanis et al., 2008***; ***Nguyen et al., 2007b***; ***Nguyen et al., 2008***; ***Hippisley-Cox and Coupland, 2012***), but ignore the risks of refracture or mortality following an initial fracture.

We hypothesize that the three events— fracture, refracture, and mortality— are correlated and that the correlation is underlined by advancing age and low bone mineral density (BMD). The present study sought to test that hypothesis by pursuing two specific aims: (i) to quantify the risk of fracture-related consequences, including refracture and post-fracture mortality; and (ii) to define the contributions of age and low bone BMD to the transition between fracture, refracture, and mortality for an individual. Addressing the aims will advance the risk assessment and help identify individuals who do badly after an initial fracture for appropriate intervention and reduce mortality burden in the general population.

## Results

### Baseline characteristics of participants

The study included 2046 women and 1205 men, all aged 60 years and older at baseline. Mean age at baseline was 70 (SD 7) in women and 70 (SD 6) in men. At baseline, approximately 22% (n = 458) women and 9% (n = 111) men were having osteoporosis (femoral neck BMD T-score $\leq$ –2.5) (*Table 1*). Prior fracture (those occurring prior to study entry) was reported in 17.5% of women and 11.5% of men. More women (38%, n = 776) than men (26%, n = 317) self-reported a fall over the previous 12 months. Men were more susceptible than women to cardiovascular disease and type 2 diabetes.

On average, the first fracture occurred at age 79 (SD 8), followed by the second fracture at age 82 (SD 7) and the third fracture at age 83 (SD 7; *Supplementary file 1*). Age at death, including death following a fracture and death without a fracture, was 83 (SD 8). Comparison between men and women reveals that despite the age at entry and at initial fracture was not substantially different, age at subsequent events, that is, second and third fracture and death, was younger in men than in women, suggesting that once a fracture occurred, men transitioned more quickly to adverse stages than women did (*Supplementary file 1*).

### Incidence of fracture and mortality

During the follow-up period, 632 (31%) women and 184 (15%) men who had sustained at least one fracture over 21,723 and 11,968 person-years were at risk, yielding a fracture incidence rate of 24 (95% CI, 22–26) and 15 (95% CI, 13–17) per 1000 person-years for women and men, respectively

**Table 1.** Baseline characteristics and incident illnesses of 2046 women and 1205 men in the Dubbo Osteoporosis Epidemiology Study.

|  | Women | Men | *P*-value |
|---|---|---|---|
| Number of participants | 2046 | 1205 |  |
| BMI (kg/m$^2$) | 26.5 (5.1) | 26.8 (3.9) | 0.035 |
| FNBMD (g/cm$^2$) | 0.81 (0.14) | 0.92 (0.15) | <0.001 |
| FNBMD T-score | −1.62 (−1.15) | −0.99 (−1.22) | <0.001 |
| Osteoporosis (n; %) | 458 (22.4) | 111 (9.2) | <0.001 |
| History of falls (n; %) | 776 (37.9) | 317 (26.3) | <0.001 |
| Prior fracture since age 50 (n; %) | 357 (17.5) | 139 (11.5) | <0.001 |
| Cardiovascular disease (n; %) | 632 (30.9) | 470 (39.0) | <0.001 |
| Bone-unrelated cancer (n; %) | 174 (8.5) | 108 (9.0) | 0.654 |
| Neurological disease (n; %) | 142 (6.9) | 69 (5.7) | 0.175 |
| Rheumatoid arthritis (n; %) | 84 (4.1) | 25 (2.1) | 0.002 |
| Respiratory disease (n; %) | 221 (10.8) | 141 (11.7) | 0.431 |
| Type 2 diabetes (n; %) | 219 (10.7) | 158 (13.1) | 0.038 |

Notes: Values shown are mean and standard deviation (in brackets), unless otherwise specified. *P*-values were derived from t-test for continuous variables and from chi-squared test for binary variables. FNBMD, femoral neck bone mineral density.

(*Figure 1*). Among the 816 individuals with fracture, 270 went on to have another fracture, with 99 having more than two fractures.

Overall, 627 (31%) women and 501 (42%) men had died over the same period, yielding a mortality rate of 33 (95% CI, 31–35) and 42 (95% CI, 38–46) per 1000 person-years for women and men, respectively. Among the deceased, 262 (42%) women and 105 (57%) men died following a fragility fracture (*Figure 1*, *Supplementary file 2*). A more detailed description of transition between health states during the follow-up period is shown in *Supplementary file 2*.

## Risk of transition between health states

The instantaneous risk of transition to next heath states is shown in *Table 2*. In fracture-free women, the instantaneous risk of having the first fracture was 2.7% (95% CI, 2.4–3.0%). Once the initial fracture occurred, the risk of sustaining another fracture was almost doubled (4.8%; 95% CI, 3.8–6.3%). This second fracture kept signaling an increased risk of further fractures. This trend was also observed for mortality: while women with no fracture had the lowest risk of mortality (1.8%; 95% CI, 1.5–2.0%), those with one, two, and three or more fractures had an increased risk of mortality from 2.1% to 12.9%.

The risk of first incident fracture was lower in men than in women (HR 0.63; 95% CI, 0.53–0.75). However, there was no significant difference in the risk of second fracture between genders. More interestingly, the risk of third fracture in men was 2.1-fold higher than that in women (HR 2.11; 95% CI, 1.13–3.95). Moreover, the risk of death, regardless of fracture status, was consistently higher in men than in women (*Table 2*).

## Risk factors for transition between states

Risk factors for the transition from no fracture to an initial fracture, no fracture to death, first fracture to second fracture, and first fracture to death are shown in *Figure 2A*. As previously known, men were less likely than women to suffer an initial fracture (HR 0.63; 95% CI, 0.53–0.75), but after a

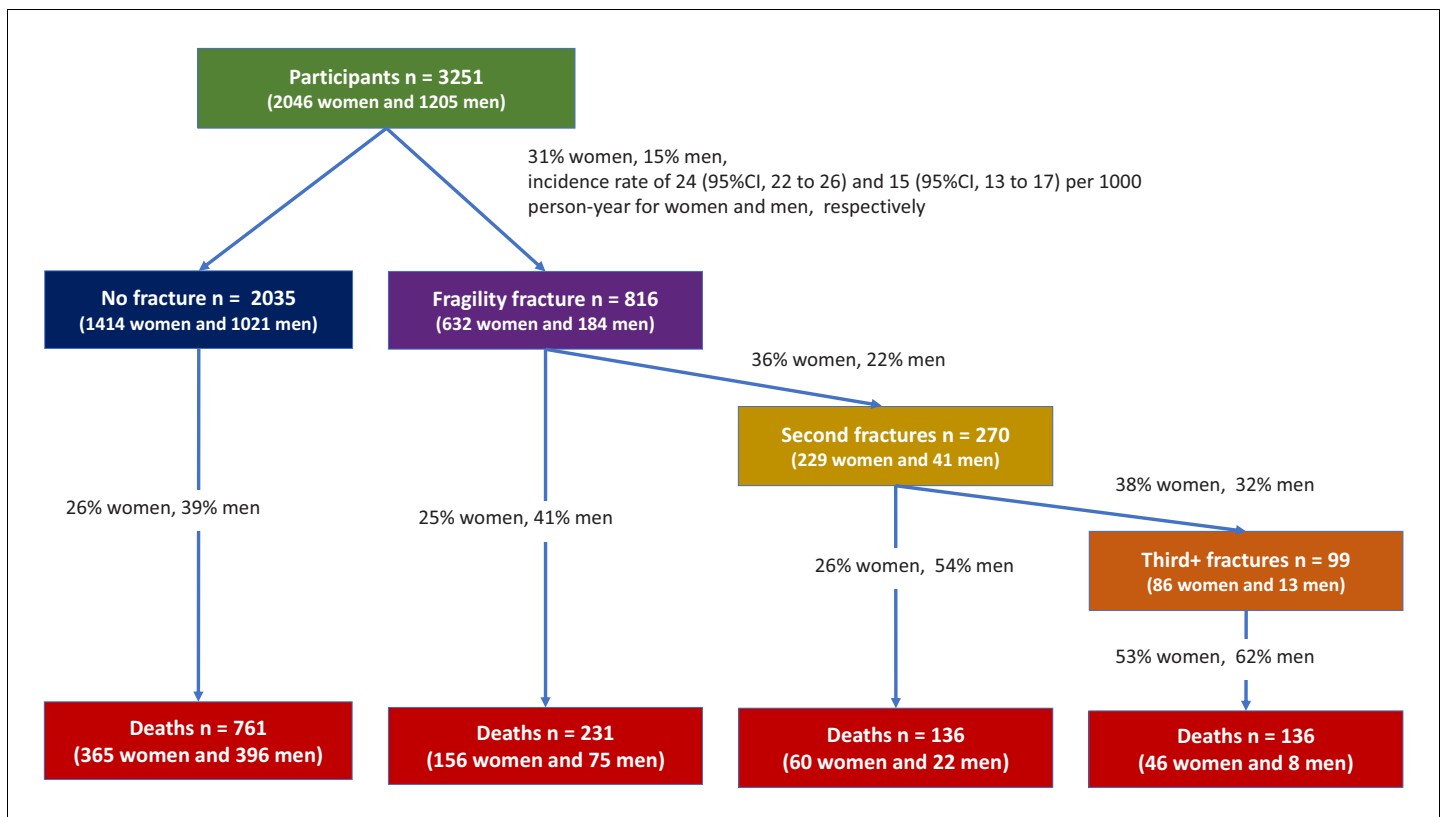

**Figure 1.** Flowchart of recruitment and follow-up.

**Table 2.** Instantaneous risk of transition between states of bone health for women and men.

| Transitional state | Women: risk (95% CI) | Men: risk (95% CI) | Hazard ratio for men vs women (95% CI) |
|---|---|---|---|
| No Fx → Initial Fx | 2.7 (2.4–3.0) | 1.7 (1.4–2.0) | 0.63 (0.53–0.75) |
| Initial Fx → Second Fx | 4.8 (3.8–6.3) | 4.1 (2.8–5.9) | 0.85 (0.61–1.19) |
| Second Fx → Third+ Fx | 6.3 (3.7–10.9) | 13.3 (6.7–26.1) | 2.11 (1.13–3.95) |
| No Fx → Death | 1.8 (1.5–2.0) | 3.2 (2.8–3.7) | 1.81 (1.55–2.10) |
| Initial Fx → Death | 2.1 (1.5–2.9) | 5.0 (3.5–7.4) | 2.40 (1.79–3.21) |
| Second Fx → Death | 2.2 (1.1–4.4) | 16.5 (8.8–32.9) | 7.52 (4.33–13.07) |
| Third+ Fx → Death | 12.9 (5.8–28.9) | 33.9 (11.8–86.2) | 2.62 (1.17–5.87) |

Note: 'Fx', fracture. CI, confidence interval. BMD, bone mineral density. BMI, body mass index. See Data Analysis for the definition of 'instantaneous risk' (hazard). Risk was estimated for a 'typical' man or woman having average BMD, characterized by mean values of predictors as follows: age at event = 70 years, femoral neck bone mineral density T-score = −1.5 (equal to mean), BMI = 26.6 kg/m$^2$ (equal to mean), no history of falls, no prior fracture, no comorbidities. Hazard ratio and 95% confidence interval were derived from the multistate model, adjusting for age, femoral neck BMD, BMI, history of a fall within 12 months, prior fracture, and other comorbidities (cardiovascular disease, cancer, type 2 diabetes, neuromuscular, rheumatoid arthritis, and chronic obstructive pulmonary disease). Bold-face values indicate a statistically significant difference between men and women. In each cell, values are percentage of risk and 95% confident interval (in the brackets).

fracture men were more likely than women to die (HR 2.40; 95% CI, 1.79–3.21) (*Figure 2B*). In either men or women, advancing age was associated with an increased risk of initial fracture, second fracture, and mortality. For a given age and gender, individuals with lower femoral neck BMD were associated with increased risks of initial fracture, second fracture, and mortality. A personal history of fracture was a risk factor for subsequent fracture, but was not associated with the transition between fracture and mortality. In both women and men, those having rheumatoid arthritis were more likely to have an increased risk of initial fracture and second fracture (*Figure 2A*).

The 'sojourn time' for each transitional status is shown in *Table 3*. Women tended to stay at each health state longer than men. The predicted sojourn time in state 1 (fracture free) of a 'typical' woman with osteopenic BMD (T-score = −1.5) was 22.4 years, 5.5 years longer than that of an osteoporotic woman (T-score = −2.5). Once the initial fracture occurred, the difference in sojourn time between a person with low BMD and a person with normal BMD was not as much as in those who have not yet sustained a fracture, especially for men.

## Individualization of risk

Using the risk factors and instantaneous risk, we estimated the 5-year probability of transition between health states for a 'typical' individual based on the individual's risk profile (*Table 4*). For a woman aged 70 years, with a BMD T-score = −1.5 and BMI being 26.6 kg/m$^2$, no history of fall, no prior fracture, no comorbidities, the probability of transition from no fracture to fracture (10.1%) was not much different from the risk of mortality (8.6%). However, once a fracture has occurred, her risk of next fracture increased by almost 1.7-fold (16.5%), which was greater than the risk of mortality (10.4%). In the same state, a woman with low BMD would have a higher risk of progressing to fracture or death than a woman with normal BMD (*Supplementary file 3*).

For a man with a similar profile as a woman, the risk of an initial fracture (6.0%) was lower than the risk of mortality (15.3%). If the man has sustained a fracture, then his risk of mortality is predicted to increase to 26.3%.

For both men and women, the risk of mortality was associated with a lower BMD T-score and advancing age (*Figure 3*). Importantly, the mortality risk increased with the increasing number of fractures; however, the increase was more pronounced in men than in women. Less than 50% of men who sustained a refracture survived longer than 5 years. *Figure 3—figure supplement 1* further illustrates the effect of BMD on the compound risk of mortality associated with the number of fractures.

The prediction model has good calibration as there was a close agreement between the observed and predicted incidence of fracture and mortality (*Figure 3—figure supplement 2*).

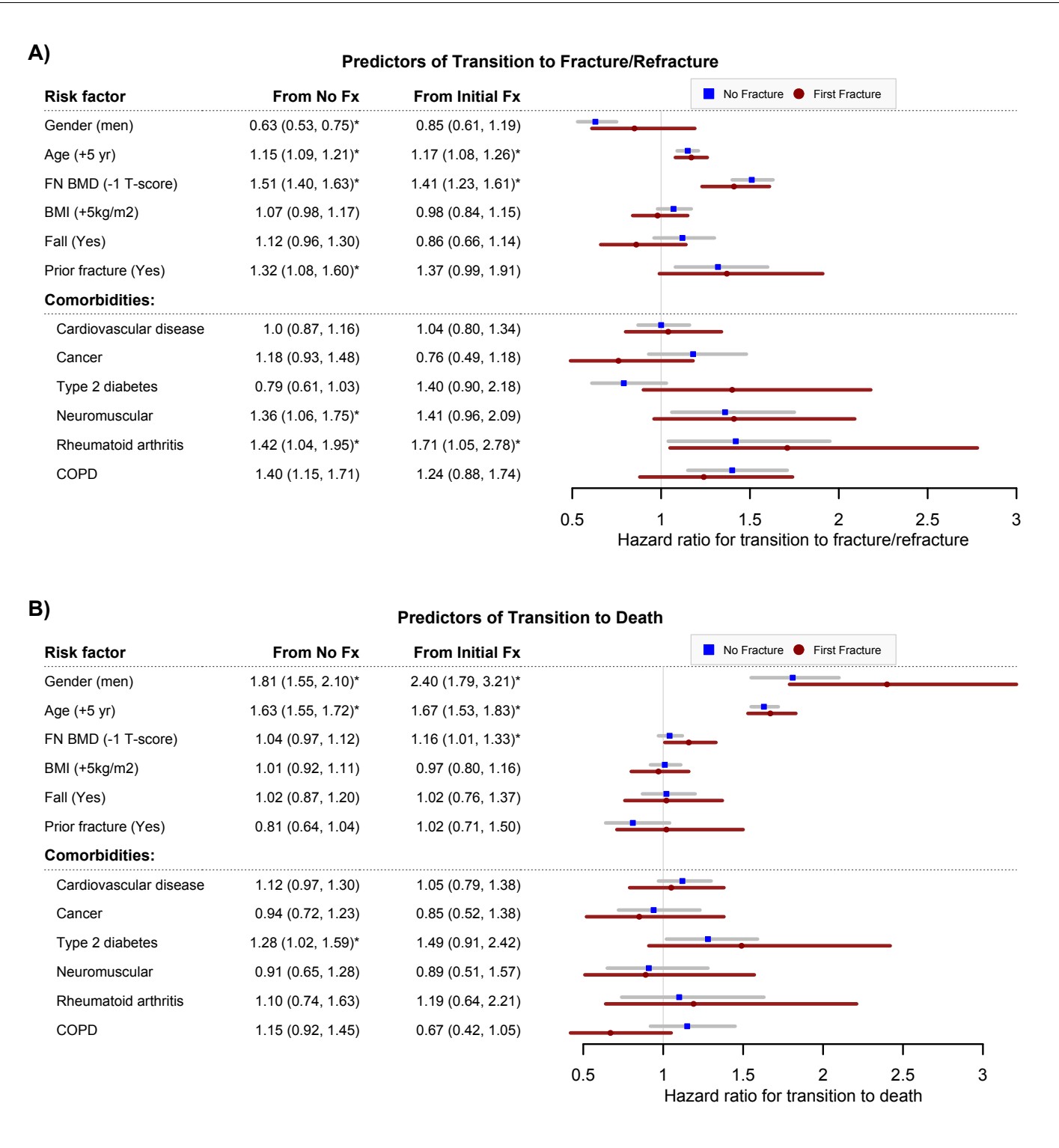

**Figure 2.** Predictors of transition to fracture/refracture (Panel **A**) and predictors of transition to death (Panel **B**): hazard ratio and 95% confidence interval from the multistate model, adjusting for age, femoral neck BMD, BMI, history of fall within 12 months, prior fracture, and other comorbidities (cardiovascular disease, cancer, type 2 diabetes, neuromuscular, rheumatoid arthritis, and chronic obstructive pulmonary disease). Fx, fracture. FNBMD, femoral neck bone mineral density. COPD, chronic obstructive pulmonary disease. Symbol * indicates statistical significance at level of 5% (p<0.05).

**Table 3.** Sojourn time (in years) of women/men with different bone health*.

| Transition | T-score = 0 | T-score = −1.5 | T-score = −2.5 |
|---|---|---|---|
| Women | | | |
| No Fx. | 32.1 (28.2–36.6) | 22.4 (20.4–24.6) | 16.9 (15.2–18.7) |
| Initial Fx. | 21.9 (16.7–28.8) | 14.5 (11.8–17.7) | 10.1 (8.8–13.4) |
| Second Fx. | 18.5 (10.1–34.0) | 11.8 (7.6–18.1) | 8.7 (5.7–13.1) |
| Third+ Fx. | 5.2 (1.7–15.9) | 7.7 (3.4–17.7) | 10.0 (4.7–21.5) |
| Men | | | |
| No Fx. | 25.5 (22.5–28.9) | 20.4 (18.3–22.8) | 16.9 (14.8–19.3) |
| Initial Fx. | 15.5 (11.5–20.9) | 11.0 (8.5–14.3) | 08.7 (6.6–11.4) |
| Second Fx. | 5.4 (3.0–9.9) | 3.3 (2.2–5.4) | 2.4 (1.5–3.9) |
| Third+ Fx. | 2.0 (0.6–7.2) | 3.0 (1.1–8.2) | 3.8 (1.5–9.9) |

*Time was estimated for women or men with different BMD profiles (i.e., 0, −1.5 vs −2.5), at the age of 70, with BMI of 26.6 kg/m$^2$ (equal to mean), with no history of falls, no prior fracture and no comorbidities. Sojourn time is defined as the predicted time an individual stays in one state before moving to the next state. Fx, fracture. In each cell, values are number of years and 95% confident interval (in the brackets).

## Discussion

In many individuals, fracture, refracture, and death were sequentially linked events: individuals with an initial fracture have an increased risk of subsequent fracture and mortality. Previous studies investigated risk factors for each pair of consecutive states at a time (*Bliuc et al., 2009*; *Bliuc et al., 2013*). This study took a systemic approach to examine these linked events in its flow in each

**Table 4.** Five-year probability of transition between states of bone health for women and men.

**Women**

| From | To | | | | |
|---|---|---|---|---|---|
| | **No fracture** | **1st fracture** | **2nd fracture** | **3rd fracture** | **Death** |
| No fracture | 80.0 (78.2–81.4) | 10.1 (9.1–11.3) | 1.2 (0.9–1.5) | 0.1 (0.1–0.2) | 8.6 (7.6–9.7) |
| 1st fracture | | 70.8 (65.3–75.4) | 16.5 (12.9–20.6) | 2.4 (1.4–3.9) | 10.4 (7.9–14.0) |
| 2nd fracture | | | 65.3 (51.4–75.5) | 18.5 (10.2–28.9) | 16.2 (10.0–26.4) |
| 3rd fracture | | | | 52.4 (22.9–75.0) | 47.6 (25.0–77.1) |

**Men**

| From | To | | | | |
|---|---|---|---|---|---|
| | **No fracture** | **1st fracture** | **2nd fracture** | **3rd fracture** | **Death** |
| No fracture | 78.3 (76.1–80.3) | 6.0 (5.0–7.0) | 0.4 (0.3–0.6) | 0.1 (0.0–0.2) | 15.3 (13.6–17.2) |
| 1st fracture | | 63.5 (54.9–70.2) | 8.1 (4.9–12.1) | 2.1 (0.8–4.3) | 26.3 (20.0–34.3) |
| 2nd fracture | | | 22.4 (8.4–38.4) | 13.6 (3.7–29.7) | 64.0 (44.5–82.5) |
| 3rd fracture | | | | 18.4 (0.8–53.8) | 81.6 (46.2–99.2) |

Note: Risk was estimated for a 'typical' man or woman having a risk profile characterized by mean values of predictors as follows: age at event = 70 years, femoral neck bone mineral density T-score of −1.5, BMI = 26.6 kg/m$^2$, no history of falls, no prior fracture, no comorbidities. In each cell, values are percentage of risk and 95% confident interval (in the brackets).

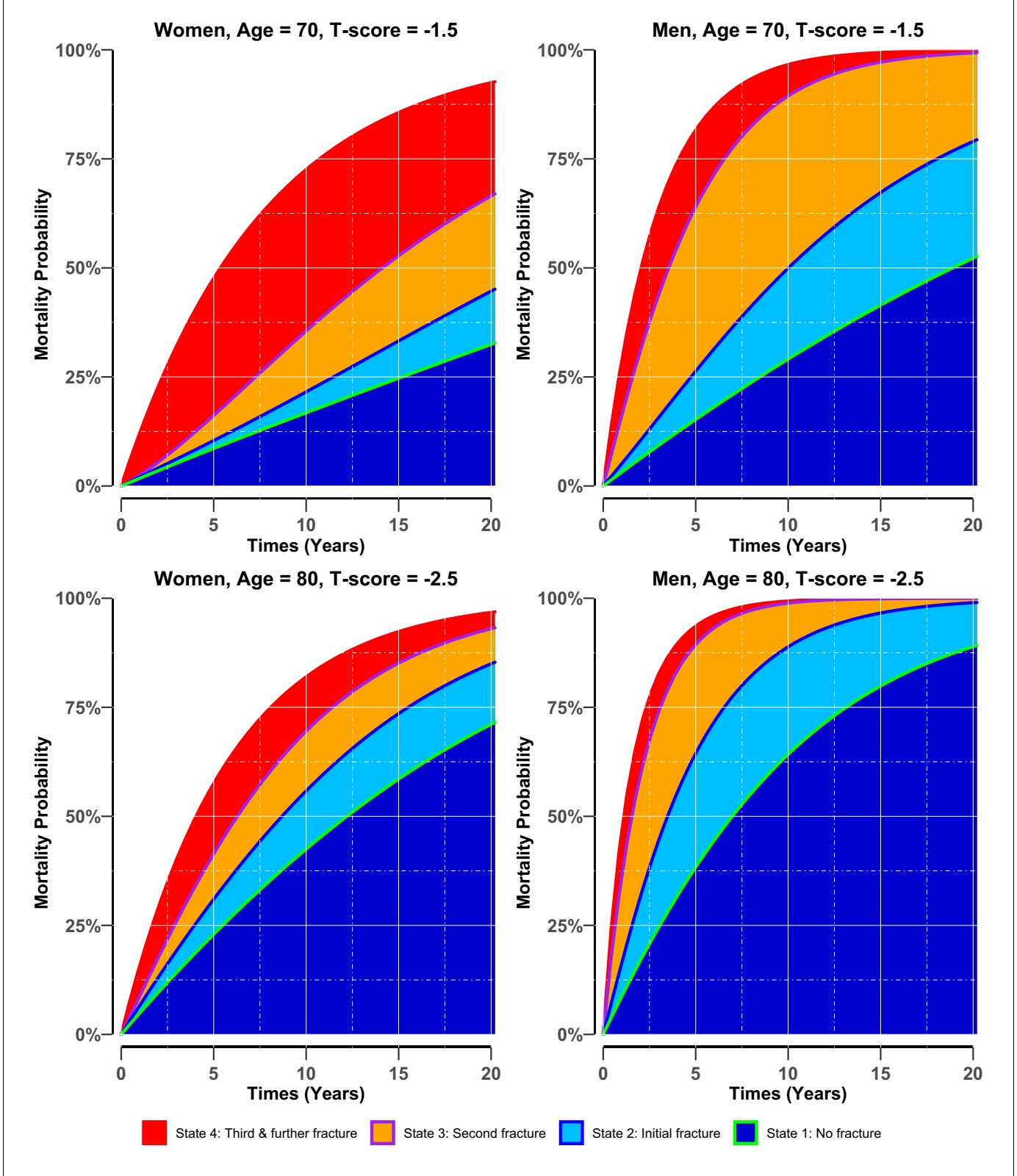

**Figure 3.** Adjusted cumulative probability of mortality in women (left panel) and men (right panel) who had stayed in different states of bone health. There were four potential bone heath states before transiting to state 5 (i.e., mortality): *state 1*: no fracture (green blue colour area) if the individual entered the study without any osteoporotic fracture; *state 2*: initial fracture (light blue area) if an individual had sustained a fracture after study entry; *state 3*: second fracture (purple-orange area) if an individual had suffered a second fracture; and *state 4*: third and further fractures (red area) if an

*Figure 3 continued on next page*

*Figure 3 continued*

individual had suffered two or more subsequent fractures during the follow-up period. Risk was estimated for women and men with different BMD profiles (i.e., −1.5 vs −2.5), at the event age of 70 and 80, having all other factors set to the population mean, that is, body mass index = 26.6 kg/m$^2$, no history of fall at baseline, no prior fracture and no comorbidities.

The online version of this article includes the following source data and figure supplement(s) for figure 3:

**Source data 1.** Adjusted cumulative probability of mortality in women and men, who had stayed in different states of bone health.

**Figure supplement 1.** Adjusted cumulative mortality probability in women (left panel) and men (right panel) by bone health state.

**Figure supplement 1—source data 1.** Adjusted cumulative mortality probability in women and men by bone health state.

**Figure supplement 2.** Assessment of goodness-of-fit of the analysis model for each of five heath states: alive and free of fracture, initial fracture, second fracture, third and further fracture, and death.

**Figure supplement 2—source data 1.** Assessment of goodness-of-fit of the analysis model: Observed and expected prevalences of each heath state.

individual, and then modeled the transition between health states. The novel outcome of this study is an individualized predictive model to predict not only the probability but also the time of an incident fracture. Moreover, the model at the same time provides these estimates for consequences of fracture, that is, recurrent fracture and premature death. More importantly, by timing the duration that people on a specific state occupy this state, for the first time, we can quantify the number of healthy years lost (in the view of bone health) due to osteoporotic fracture and recurrent fracture.

This is, to our knowledge, the first investigation into the transition between fracture and fracture-associated events. However, for each event our findings were similar to those from previous studies, in that women have a higher risk of fracture than men (*Johnell and Kanis, 2005*) and once men sustain the first fracture, risk of the second fracture is similar to that in women (*Center et al., 2007*). Moreover, in contrast to risk of fracture, risk of death, regardless of fracture status, is greater in men than in women (*Bliuc and Center, 2016*; *Kannegaard et al., 2010*). However, the difference in risk estimation in our study and previous studies is that whereas many other studies reported lifetime risk (*Johnell and Kanis, 2005*), our model estimated instantaneous risk, beside with the estimation of 5-year risk. The estimation of lifetime risk for fracture can be misleading as once an incident fracture, an event with several consequences, occurs, this event shifts the remaining lifetime risk of the individual.

The progression to premature mortality following a fragility fracture has been described in many studies, but the underlying mechanism is still unclear. Mortality following a hip fracture is best studied due to its severity. Adverse events related to surgery to repair a fractured hip have been also implicated in the increased mortality observed among the older people (*Nikkel et al., 2015*). However, the specific cause for long-term increase in mortality following hip fracture and other types of fracture is largely unknown. The role of comorbidities has been reported but with inconsistent findings across studies (*Cenzer et al., 2016*; *Cree et al., 2000*; *Liem et al., 2013*). Risk factors for fracture such as low bone mineral density, bone loss, and low muscle strength have recently been linked to mortality risk in the general population as well as post-fracture mortality (*Nguyen et al., 2007c*; *Van Der Klift et al., 2002*; *Kado et al., 2000*; *Rantanen et al., 2000*; *Pham et al., 2017*).

We found that once the initial fracture occurred, the difference in sojourn time between a person with low BMD and a person with normal BMD was not as much as in those who have not yet sustained a fracture, especially for men. This suggests that after an initial fracture, factors other than BMD might play a more important role than BMD in the progression to subsequent fractures and premature death. Therefore, further studies to investigate which factors are dominant of the progression after an initial fracture are required. The shorter transition time after an initial fracture also suggests that any intervention strategy which focuses on improving BMD would be more beneficial if implemented at early stage than at later stages.

Current tools for fracture prediction suffer from a number of major weaknesses (*Nguyen and Eisman, 2018*): lack of mortality data and no contextualised estimate of risk. All existing prediction models such as the Garvan Fracture Risk Calculator (*Nguyen et al., 2008*) and FRAX (*Kanis et al., 2008*) provide only an estimate of fracture risk, with no estimate of mortality risk. This is a weakness because mortality is clearly strongly associated with fracture (*Cree et al., 2000*; *Bliuc et al., 2009*; *Nguyen et al., 2007c*) and treatment can reduce fracture-associated mortality risk (*Lyles et al., 2007*; *Reid et al., 2018*). Moreover, the risk estimate produced by these prediction models are not put into context. They do not provide the benefit in terms of fracture reduction and

increased survival (and potential risk) if a high-risk patient opts for treatment; limiting the communication of risk and clinically useful discussions between patients and their physicians.

Thus, our results have important implications for fracture risk prediction and risk communication. Unlike other chronic diseases where the deterioration of the functional organs is associated with clinical signs, the deterioration of bone health is mainly silent until the first fracture occurs. However, the lack of adverse events estimation in the existing fracture risk prediction tools can be an obstacle for both patients and doctors to be fully aware of the significance of the patients' bone health condition, which would lead to under-management of the conditions. By providing the estimate of fracture risk and the estimate of mortality risk, our unified model will enable patients and doctors to fully appreciate the serious nature of fragility fracture. Patients do not always appreciate the serious consequence of fracture (e.g., subsequent fractures and mortality), and as a result, they usually underestimate their risk of adverse outcomes. Patients are however concerned about quality of life and mortality, and providing the estimated risk of mortality can motivate them to take preventive measures.

One way to convey the new compound risks of fracture and mortality is to transform the risks into 'skeletal age'. Based on the idea of 'lung age' (*Morris and Temple, 1985*) and 'effective age' (*Spiegelhalter, 2016*), skeletal age can be defined as the age of an individual's skeleton as a result of the individual's risk factors for fracture. In the normal circumstance, skeletal age is the same as chronological age, but in high-risk individuals, skeletal age is greater than chronological age. The number of years lost or gained in effective age for a risk factor with mortality hazard ratio of H is $\log(H)/\log(h)$, where $h$ represents the annual risk of mortality which is approximately 1.1 (*Spiegelhalter, 2020*). For instance, for a 70-year-old man who has sustained a fracture, the hazard ratio of 1.67 (*Figure 2B*) is equivalent to a loss of ~ 5.4 years of life ($\log(1.67)/\log(1.1)$). In other words, for the 70-year-old man, the hazard ratio of 1.67 corresponds to a skeletal age of 75.4 years. In other words, if an individual is 70 years old (chronological age) but skeletal age is 75.4, then this means that the individual is the same risk profile as a 75.4-year-old individual with a 'healthy profile.' Moreover, for an individual who has sustained a fracture, each standard deviation lower in femoral neck BMD on average take around 1.5 years of life of the individual.

Evidence from randomized controlled trials suggest that in patients with osteoporosis and/or a pre-existing fracture, bisphosphonate treatment reduces fracture risk and mortality risk. In a placebo-controlled trial, Lyles and colleagues found that among elderly hip fracture patients, intravenous zoledronic acid reduced the risk of subsequent fracture by 35% and reduced mortality risk by 28%, regardless of bone mineral density (*Lyles et al., 2007*). A recent placebo-controlled trial further showed that in osteopenic patients with a fracture, zoledronic acid also reduced mortality risk with an odds ratio of 0.65 (95% CI, 0.40–1.05) (*Reid et al., 2018*). A review of all clinical trials reported that treatment of osteoporotic patients with medications with proven fracture efficacy reduced mortality risk by approximately 10% (*Bolland et al., 2010*). However, a recent meta-analysis found that the effect of bisphosphonate treatment on mortality was less certain (*Cummings et al., 2019*). Taken together, these results suggest that in osteoporotic or osteopenic patients with a fracture, bisphosphonate treatment may reduce mortality risk. The beneficial effect of treatment can also be expressed in terms of 'skeletal age'.

Our results clearly show that the risk of fracture and subsequent events should be individualized. This is true, because there is no 'average person' in the population. Two women having the same BMD and age but different fracture history could have different risk estimates, and this difference must be taken into account in the assessment of risk. Our new model provides a tool and a framework for including other risk factors such as genetic profile (*Ho-Le et al., 2017*; *Ho-Le et al., 2021*; *Nielson et al., 2016*) and bone microarchitecture (*Karasik et al., 2017*; *Pepe et al., 2016*) to be included in the personalized assessment of fracture and fracture-related outcomes. An important advantage of using genomic data in fracture risk assessment is that genotypes do not change with time, and as a result, the risk of fracture for the individual can be predicted at younger ages, well before the conventional risk factors become apparent. Although there is no 'genetic therapy' for individuals at high risk of fracture, the use of an osteogenomic profile could help segregate individuals at high risk from those with low risk of fracture, and facilitate educational aspects of prevention and counseling services.

These findings should be interpreted within the context of strengths and weaknesses. The strength of the study is its prospective design and the long follow-up of 21 years allowing us to

identify a large number of multiple subsequent fractures and death, therefore, making it possible for the transition analysis. However, the cause of death was not available, and it was not possible to conduct an in-depth analysis of mortality-attributable risk. Because this cohort included mainly Caucasians (98.6%) (*Nguyen et al., 2008*), the present findings might not be generalizable to other ethnicities. Despite the relative large sample size in general, the number of men who sustained three and more fractures was not sufficiently enough for the statistical analysis to produce a stable result for this group. In the present study, the transition between states of bone health following specific fracture types has not been investigated due to the modest number of events in each group of type of fracture. Comorbidities occurring during the follow-up time were not ascertained and could not be treated as time-variant covariates in the analysis.

In summary, we have developed a multistate model that provides the general community with not just fracture risk estimate but also the likelihood of refracture and survival. This information can encourage at-risk people to proactively make changes in lifestyle to mitigate their elevated risk. Our model also provides a personalized window of opportunity for intervention to reduce the burden of fracture-associated outcomes in at-risk individuals.

## Materials and methods

### Key resources table

| Reagent type (species) or resource | Designation | Source or reference | Identifiers | Additional information |
| --- | --- | --- | --- | --- |
| Software, algorithm | R Project for Statistical Computing | R Project for Statistical Computing | RRID:SCR_001905 | |
| Software, algorithm | Algorithm SAS program | Statistical Analysis System | RRID:SCR_008567 | |

### Participants and setting

This study was part of the ongoing Dubbo Osteoporosis Epidemiology Study (DOES) which was designed as a population-based prospective investigation, with the setting being Dubbo and surrounding districts. The study was commenced in 1989, and is still ongoing. Dubbo was selected for the study site because it has a relatively stable population whose age structure resembled that of the Australian population at large. Moreover, because Dubbo city has only two radiological services that can cover the totality of fracture ascertainment for local residents. In 1989, approximately 2100 women and 1600 men aged 60 years or over were living in the city of Dubbo (*Jones et al., 1994*), and at the time of commencement (1989), DOES involved over 60% of this population, with 98.6% of them being Caucasian origin (*Bliuc et al., 2015*). The median follow-up time for the cohort was 9 years (interquartile range 5–18 years). The study was approved by the Ethics Committee of St Vincent's Hospital (Sydney) (HREC reference number 13/254) and carried out according to the Australian National Health and Medical Research Council (NHMRC) Guidelines, consistent with the Declaration of Helsinki (established in 1964 and revised in 1989) and US Food and Drug Administration guidelines. All participants have provided written informed consent.

At the time of conception (mid-1989), DOES had hypothesized that the risk of fracture could be predicted by about 10 risk factors, and under the presumption that (i) each factor needs at least 10 events, (ii) the incidence rate is about 1% per year, then a sample of 2000 individuals would be required to follow for 5 years. Ultimately, the study has recruited more than 3200 individuals of both genders, and over the past 20 years, the number of fractures was 632 in women and 184 in men. In other words, the number of events is adequate for developing a reproducible prediction model.

### Fracture ascertainment

Fractures occurring during the study period were identified through radiologists' reports from the only two radiology centers providing x-ray services within the Dubbo region as previously described (*Nguyen et al., 2007d*). Circumstances surrounding the fracture were also confirmed with participants on the next visit that was close to the fracture. In this study, we included only fractures having definite reports and resulting from low-energy trauma such as falls from standing height or less in this analysis. Fractures due to malignant diseases or high-impact trauma (e.g., motor vehicle accident, sport injury, or fall from above standing height) were excluded from the analysis. No systematic x-ray screening for asymptomatic vertebral fracture was conducted; therefore, vertebral

fractures were incidental findings in x-ray reports or were x-rayed due to the presence of back pain. Fractures of the skull, fingers, and toes were not included in the analysis.

The incidence of death was ascertained by the NSW Registry of Births, Deaths and Marriages. Deaths were also monitored by systematically searching funeral director lists, local newspapers, and on radio, or by word of mouth with a confirmation or biannual telephone contact.

## Bone measurements and risk factor assessment

Bone mineral density (BMD) was measured at the femoral neck and lumbar spine by dual energy x-ray absorptiometry (DXA), using GE LUNAR DPX-L and later PRODIGY densitometer (GE LUNAR, Madison, WI). The radiation dose used is less than 0.1 µGy and the coefficient of variation of BMD at our laboratory is 0.98% for lumbar spine and 0.96% for femoral neck (*Nguyen et al., 1997*; *Ho-Le et al., 2021*). The femoral neck BMD was converted into T-score using our own young population reference values of mean and standard deviation; mean of 1.00 (SD 0.12) is for women and 1.04 (SD 0.12) is for men. Approximately 1.5% of participants had no femoral neck BMD measurement at baseline; and those missing values were imputed based on age, height, weight, and lumbar spine BMD at baseline, using the multivariate imputation by chained equations algorithm (*van Buuren and Groothuis-Oudshoorn, 2011*).

Body weight (kg) was measured in light clothing and without shoes using an electronic scale. Height (cm) was measured without shoes by a wall-mounted stadiometer. Body mass index (BMI, kg/m$^2$) was calculated based on the weight and height measured at baseline. History of falls was obtained via a structured questionnaire administered by a trained nurse during the interviews at baseline and biennial follow-up visits.

## Data analysis

A multistate model was used to describe the transition between a series of states in continuous time for an individual. We considered five states: (1) an individual at state 1 if the individual entered the study without any fracture; (2) state 2 if an individual had sustained a fracture (after study entry); (3) state 3 if an individual had suffered a second fracture; (4) state 4 if an individual had suffered two or more subsequent fractures during the follow-up period; and (5) state 5 if an individual had died during the follow-up period. Any individual at state 1, 2, 3, or 4 could transit to state 5 (i.e., death). Once individuals entered state 5 (i.e., they could no longer move to any other state); therefore, death is called '*absorbing state*' (*Figure 4*).

In the multistate model, the transition from state $r$ to $s$ is governed by *transition intensities* ($q_{rs}$) which are estimated from the observed data using the maximum likelihood method. These transition intensities represent the instantaneous risk of moving from current state (state $r$) to the next state (state $s$). Instantaneous risk (also referred to as 'hazard') is the probability that an individual would experience an event at a particular given point in time. This risk is very small over a very short time period, but can accumulate over time. Thus, the incidence of an event is the instantaneous risk multiplied by the length of time. In our study, except for the absorbing state, for states 1 to 4, at any time point, individuals who stay in one of these states have an instantaneous risk of staying in that state ($q_{rr}$) and an instantaneous risk of moving to the next states ($-q_{rs}$) (*Figure 4*). At a given time point, for example, a fracture-free individual (state 1) had an instantaneous risk of staying at that state 1 ($q_{11}$), an instantaneous risk of suffering an initial fracture ($q_{12}$), and an instantaneous risk of death ($q_{15}$). These instantaneous risks were estimated for an individual based on the individual's risk profile and adjusted for age. From these instantaneous risks, we can quantify the individualised risk of fracture, refracture, and mortality for an individual. Besides transition intensities, we also estimated the *sojourn time* which is defined as the predicted time an individual stays in one state before moving to the next state.

The effect of each potential risk factor on transition intensities was estimated as relative risk. Our model considered six specific risk factors: age, bone mineral density, body mass index, a history of falls, prior fracture, and common diseases. Age was included in the model as a time-variant factor, and for simplicity, age henceforth is referred to as *age at event*. Age at event is the age of an individual at the beginning of each state. The reason to adjust for age at event rather than age at baseline is to avoid the problem of immortal time bias. Immortal time bias could occur in groups where individuals transitioned through several states as they had to survive long enough to be able to

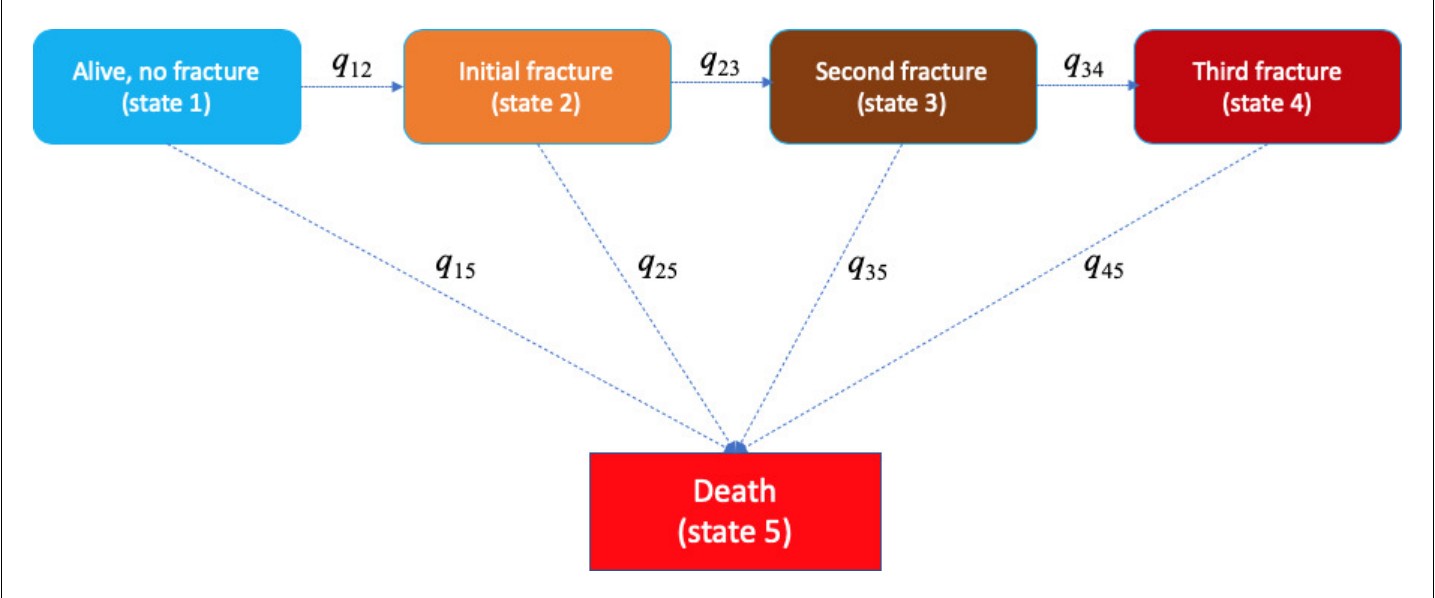

**Figure 4.** Markovian model of transition between five heath states (e.g., alive without a fracture, initial fracture, second fracture, third and further fractures, and death). The transition between state *r* to state *s* is 'governed' by the instantaneous risk of transition (i.e., intensity) $q_{rs}$, where *r* and *s* represent each of the five health states. At a given time point, for example, a fracture-free individual (state 1) had an instantaneous risk of staying at that state 1 ($q_{11}$), an instantaneous risks of suffering an initial fracture ($q_{12}$), and an instantaneous risk of death ($q_{15}$).

make these transitions. Other covariates were treated as time-invariant factors. All data analysis and modeling were conducted with SAS software version 9.4 (SAS Institute, Inc Cary, NC, USA) and R Statistical Environment (*R Development Core Team, 2008*). The multistate model was fitted with the R package *msm* (*Jackson, 2011*).

## Acknowledgements

The authors gratefully acknowledge the expert assistance of Janet Watters, Donna Reeves, Shaye Field, and Jodie Rattey in the interview, data collection, and measurement of bone densitometry, and the invaluable help of the Dubbo Base Hospital radiology staff, PRP Radiology and Orana radiology. We thank the IT group of the Garvan Institute of Medical Research for help in managing the data. This work is supported by NHMRC and in part by a grant from the Amgen Competitive Grant Program (2019).

## Additional information

### Competing interests

Jacqueline R Center: has given educational talks for and received travel expenses from Amgen, Merck Sharp & Dohme, Novartis, Sanofi-Aventis. She has received travel expenses from Merck Sharp & Dohme, Amgen and Aspen. John A Eisman: has served as consultant on Scientific Advisory Boards for Amgen, 35 Eli Lilly, Merck Sharp & Dohme, Novartis, Sanofi-Aventis, Servier and deCode. Tuan V Nguyen: has received honoraria for consulting or speaking in symposia sponsored by Merck Sharp & Dohme, Roche, Sanofi-Aventis, Novartis, Amgen, and Bridge Healthcare Pty Ltd (Vietnam). The other authors declare that no competing interests exist.

### Funding

| Funder | Grant reference number | Author |
| --- | --- | --- |
| National Health and Medical Research Council | NHMRC APP1195305 | Tuan V Nguyen |

| | | |
|---|---|---|
| Amgen | Competitive Grant Program (2019) | Tuan V Nguyen |
| Amgen | Christine & T. Jack Martin Research travel grant | Thao Phuong Ho-Le |
| Australian and New Zealand Bone and Mineral Society | Christine & T. Jack Martin Research travel grant | Thao Phuong Ho-Le |

The funders had no role in study design, data collection and interpretation, or the decision to submit the work for publication.

### Author contributions

Thao Phuong Ho-Le, Conceptualization, Data curation, Software, Formal analysis, Validation, Visualization, Methodology, Writing - original draft, Writing -review and editing; Thach S Tran, Hanh M Pham, Data curation, Formal analysis, Methodology, Writing - original draft, Writing - review and editing; Dana Bliuc, Conceptualization, Data curation, Writing - review and editing; Steven A Frost, Formal analysis, Investigation, Writing - review and editing; Jacqueline R Center, Resources, Data curation, Investigation, Writing - review and editing; John A Eisman, Conceptualization, Resources, Investigation, Project administration, Writing - review and editing; Tuan V Nguyen, Conceptualization, Resources, Data curation, Formal analysis, Supervision, Funding acquisition, Validation, Investigation, Methodology, Writing - original draft, Project administration, Writing - review and editing

### Author ORCIDs

Thao Phuong Ho-Le (iD) https://orcid.org/0000-0002-8387-1893

### Ethics

Human subjects: The study was approved by the Ethics Committee of St Vincent's Hospital (Sydney) (HREC reference number: 13/254) and carried out according to the Australian National Health and Medical Research Council (NHMRC) Guidelines, consistent with the Declaration of Helsinki (established in 1964 and revised in 1989) (US Food and Drug Administration). All participants have provided written informed consent.

### Decision letter and Author response

Decision letter https://doi.org/10.7554/eLife.61142.sa1
Author response https://doi.org/10.7554/eLife.61142.sa2

## Additional files

### Supplementary files

• Supplementary file 1. Age at study entry, initial fracture, second fracture, third fracture, and death in women and men. Values shown are mean and standard deviation (in brackets).

• Supplementary file 2. Transition between health states during the study period: actual number of individuals and probability for 2046 women and 1205 men. Data shown are the number of individuals (percentage in brackets). The sign '–' in a state indicates that the transition to the state is shown in the next row.

• Supplementary file 3. Five-year probability of transition between states of bone health for women and men with femoral neck BMD T-score of 0 (normal) and −2.5 (osteoporosis). Risk was estimated for a man or woman characterized as follows: age = 70 years, BMI = 26.6 kg/m$^2$ (equal to mean), no history of falls, no prior fracture, no comorbidities. Bold values indicate transition probability for initial and subsequent fractures significantly different between an individual with a BMD T-score of 0 (normal) and one with a BMD T-score of −2.5 (osteoporosis). In each cell, values are percentages of risks for T-score = 0 and T-score = −2.5, separated by a slash sign.

• Transparent reporting form

## Data availability

All data generated or analysed during this study are included in the manuscript and supporting files. Source data files have been provided for Figures 3, Figure 3 - figure supplement 1, and Figure 3 - figure supplement 2.

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
