## [Decision Letter]

**Acceptance summary:**

Your studies emphasize the greater risk of death with initial and subsequent fractures in an individual with low bone mass or osteoporosis. That greater risk of death not only is present with the initial fracture in individuals, but is accelerated by repeat fracture events. You also advance the concept of skeletal age, underscoring the fact that once a fracture has occurred, the condition of bone tissue has deteriorated as a composite of the risk factors for fracture represented in the individual patient.

**Decision letter after peer review:**

Thank you for submitting your article "Predicting mortality and refracture following an initial fracture" for consideration by *eLife*. Your article has been reviewed by three peer reviewers, one of whom is a member of our Board of Reviewing Editors, and the evaluation has been overseen by Clifford Rosen as the Senior Editor. The following individual involved in review of your submission has agreed to reveal their identity: Gina Woods (Reviewer #3).

The reviewers have discussed the reviews with one another and the Reviewing Editor has drafted this decision to help you prepare a revised submission.

Our expectation is that the authors will eventually carry out the additional analyses and report on how they affect the relevant conclusions either in a preprint on bioRxiv or medRxiv, or if appropriate, as a Research Advance in *eLife*, either of which would be linked to the original paper.

Summary:

This interesting paper aligns quantitatively the recurrence of fractures in a well-known cohort of aging men and women in Australia with mortality over time. The key goal of the paper is to demonstrate that men and women who fracture have greater mortality than non-fracturing individuals. While this message has noted before, the quantitative handling of the data from the cohort is the new and welcome addition to the literature. The authors look at transitions from different states of fracture and estimate instantaneous risks. Mortality has not heretofore been a component in patient-physician joint decision-making about undertaking therapy for osteoporosis.

Please address the reviewers’ comments and questions below.

Title: Please see the suggestions of reviewer 3 below for possible revision of the title itself for the paper.

Reviewer #1:

This is an interesting further analysis of the Dubbo osteoporosis study. Its headline conclusions are easily understood, though the methodology involves esoteric mathematics that will not be readily accessible to clinicians. However, it will be of interest to epidemiologists. My specific questions are as follows:

1) The concept of instantaneous risk will be unfamiliar to many readers. It needs to be explained in more detail to make these sections of the manuscript understandable.

2) Subsection “Instantaneous risk of transition between health states”, where instantaneous risks are being presented, have these calculations been adjusted for age? Those who have already had one, two or three fractures will on average be older than those with fewer fractures and so this will impact on their risk of the next fracture or of death.

3) The discussions of "typical 70 year olds" are problematic where risk at a T-score of 0 is contrasted with a risk of a T-score of -2.5. T=0 is certainly not typical of a 70 year old, where the average value is closer to -2.0.

Reviewer #2:

The manuscript "Predicting mortality and refracture following an initial fracture" by Ho-Le and colleagues reports on data from the Dubbo cohort that examines first fracture, subsequent fracture and mortality in 2046 women and 1205 men followed for 20 years. The authors report that 31% of the women and 15% of the men sustain a first fracture. The risk of the second fracture is 36% in the women and 22% of the men with a mortality that is higher in the men (41%) than in the women (25%). The key predictors of mortality are being male (HR 2.4), old (HR 1.67), and having fem neck BMD (1.16) – all with acceptable confidence intervals. The authors make a prediction model in an effort to provide an evidence-based treatment decision support. The authors feel the impact of their work will be to "change the perception of osteoporosis" because their work is pointing out the "mortality risk".

1) Abstract:

a) The authors are working on an important area in osteoporosis care – the impact of initial fractures in the elderly and of subsequent fractures on mortality. There is no question mortality is key, but many of the observations that are made and noted in the Abstract of the paper are not new. Male gender and age and BMD – all are known to correlate with poorer outcomes and recurrent fractures in osteoporosis. All the data supporting FRAX and other calculators use these variables. This is not new.

b) Please re-write or revise the sentences in your Abstract as it is not crystal clear what you mean. Is the 31% meaning 31% of the women (634 women)? Did 634 women have a fracture and over what period of time? I think you will need to state the period of observation. Was it the full 20 years? Similarly, with the men (~181 men), when did this 15% have their fractures? Was it 20 years into the collection of data or earlier? Please state this.

c) When you say the patients sustained a second fracture, what does 36% (women) and 22% (men) refer to? Is this referring back to the original group of women – i.e., 36% of the 634, and for men 22% of the 181 men? Please sharpen up the numbers and clarify the detail. The numbers may be in one of the tables, but it should be clear from the Abstract what you are talking about.

d) eTable 1 and 2: the numbers are all different from what you are presenting in the Abstract. Your Abstract needs to be understandable on its own, since most readers only read the Abstract (and then cite or not a paper).

2) When you refer to “eTable 2”, where are the numbers for death following a fragility fracture – 262 women, 105 men. Or are you saying that of the patients who died overall – 627 women, 501 men – these are the %'s of women and men (262 and 105). And the numbers mentioned are "data not shown" in the Tables?

3) eTable 3: what is "instantaneous risk" of transition? Please consider including this definition in the legend to this table. Also please explain what is in the legend. This reviewer does not believe that a general reader will understand (reading the "Note" at the bottom of the table) what was done to generate the numbers in this table. It seems like it will not be clear to the general reader why this set of assumptions were selected. How do the instantaneous risks lead to the Hazard Ratio's in the final column of Table 3 or are they unrelated variables?

4) In the presentation of the results in Figure 2A, if the reviewer is reading the data correctly you are seeing an increased risk of transition to fracture/refracture with history of RA for both men and women, which could be highlighted in the paper. The other factors do include age, FN BMD, and prior fracture which you have mentioned in the Abstract and in the subsection “Risk factors for transition between states”. Of interest, neuromuscular disease and COPD are important predictors of fracture/refracture in men and not in women. Was the statistical power (# of events in this modest sized cohort) sufficient to be definitive about the predictive value of the specific clinical risks and disorder (i.e., enough cases, etc)? This question is for Figure 2A and B.

5) What does the sentence "The predicted probability of mortality (e.g., fracture, refracture or mortality) for women and men…." Do you mean morbidity and mortality, because when you say "mortality" and then say for example – fracture, refracture or mortality – it is not clear what you mean.

6) Figure 3: this is a very important figure in your paper. It is not clear to the reviewer why you chose to do the hypothetical T score of 0 and -2.5 in the text and then chose to do T scores of +0.5 and -2.5 in the Figure 3. Please explain. Also the figure legend describes 5 bone health states but there seem to be only 4 states in the boxes (state 1,2,3,4). In the legend there is no mention of orange, there is mention of green, and there is no green apparent in the figure and there are two different blue's. There needs to be some attention given to clarifying the boxes, the legend and the figure itself and harmonizing it all. What does "event age" mean? Please define it. When you label these "age 60" vs. "age 80", do you mean this is the probability for the next 20 year for mortality (i.e., age 60-80, and ages 80-100)? If all subjects had to be 60 years old + to get into the study, there would be relatively few who were age 60 when their event (fracture) occurred from which to generate these curves. Could you provide insight into what data you had that went into the “age 60” as well as age 80? It is appreciated that these are estimates but what could you be estimating from at age 60 when you are talking about State 3 or 4? It is expected that it would take some years to get that 2nd, 3rd or 4th fracture.

7) Discussion:

a) The progression to premature mortality after a fracture could be due to the acquisition over time of other co-morbid conditions that occur in aging. Are you sure that strokes, infarct, cancer, infections/sepsis/COVID-19 etc did not occur in the years post a fracture? After reading the Materials and methods it is not clear that once a subject is enrolled that he or she would be removed from analysis if terminal cancer developed. Granted you have presumably the same rates of cancer across the board, for example, but some of these groups may not be so large – 2, 3, 4 fractures. There would be no reason to anticipate that terminal events tracked with fractures especially if the fracture was years before or tracked with the fractures. There has been some attention to this issue with hip fractures because of the sarcopenia and disability that they cause – perhaps mortality is related to that attendant frailty. However, here you are counting all fragility fractures and it is hard to see that they would have the same impact as hip fracture on the overall health.

b) In the sixth paragraph of the Discussion you are discussing estimating fracture risk and mortality risk, but you then say that patients "usually underestimate their overall health". Do you mean overestimate their health or underestimate their risk of death?

c) While you have provided models of risk of death, you have not established causality between the fractures and mortality to my review of the paper. Therefore it seems an over-strong statement to say – "…providing the estimated risk of mortality can motivate them to take preventive measures." In an non-interventional, observational study, you cannot assume that fracture prevention will significantly change the mortality risks, curves.

d) Were patients in this cohort ever given any treatments and how did you handle concomitant medications such as estrogen, bisphosphonates, SERM, testosterone and so forth? If the cohort was started in 1989 (21 years would be through 2009), and repeated fractures were occurring in hundreds of these patients, were they treated, if so with what and how was that handled in the data analysis? Were individuals specifically not treated, which would be an ethical issue? You also state (subsection “Participants and Setting”) that the study is ongoing. Surely by now subjects in the cohort have been receiving treatments for osteoporosis.

e) It would be of interest to the reader to know the breakdown of fractures that you counted. How many were clinical vertebral, hip, humerus, rib, tibia, fibula, wrist etc. What kinds of fractures are these models resting on? What are the causes of death in the cohort?

8) The reports were reviewed, but were the fractures adjudicated? Were all deaths confirmed? (Subsection “Fracture ascertainment”).

9) One wonders if Figure 1 is needed and whether the figures mainly containing the dots are needed. Please consider another shorter way to convey this information.

10) Please read the legend to eTable 4. Please revise "the person who ages of 70". There is a misspelling in the legend to Supplementary eFigure 1.

Reviewer #3:

The authors used data from the Dubbo Osteoporosis Epidemiology Study to analyze risk for initial fracture, subsequent fractures and mortality in older men and women followed for 20 years. They found that fractures were common, occurred earlier in women than in men, and were associated with earlier mortality. Mortality risk following a fracture was higher in men compared to women. The authors presented a detailed analysis of refracture and mortality rates in men and women following fracture at age 60 and age 80, to create a prediction model.

1) The major findings of this study are not new. We already know that fractures occur at an earlier age in women compared to men, and that mortality following fracture is higher in men than women. We also know that advancing age and lower BMD are associated with higher risk for fracture. This study presents a more detailed analysis of risk for subsequent fracture and mortality following a fracture than previous studies have done.

2) The author claim to have developed a prediction model to help patients and doctors make evidence-based treatment decisions. I did not find this study to offer a clinically useful risk prediction model, nor does this study address osteoporosis treatment. It does offer new information about mortality following fracture. Life expectancy is not commonly discussed in osteoporosis care, in the way it may be discussed in cancer treatment, for example. Given the average age of participants in this study, and without presenting evidence that osteoporosis treatment can influence mortality rates, I am not sure that these data will be readily incorporated into osteoporosis treatment discussions.

3) Impact Statement: "Our results will change the perception of osteoporosis because it increases mortality risk. The personalized risk prediction model presented here will change the assessment of fracture risk in the future."

Many patients are diagnosed with osteoporosis based on T-score criteria (no prior fracture). Figure 2B shows that in those without prior fracture, BMD is not associated with mortality. Perhaps this impact statement should read "osteoporotic fracture increases mortality risk".

---

## [Author Response]

Summary:This interesting paper aligns quantitatively the recurrence of fractures in a well-known cohort of aging men and women in Australia with mortality over time. The key goal of the paper is to demonstrate that men and women who fracture have greater mortality than non-fracturing individuals. While this message has noted before, the quantitative handling of the data from the cohort is the new and welcome addition to the literature. The authors look at transitions from different states of fracture and estimate instantaneous risks. Mortality has not heretofore been a component in patient-physician joint decision-making about undertaking therapy for osteoporosis.

Thank you for the summary which captures the key idea of the paper. Although the association between fracture and mortality has previously been observed by us and others, many doctors still do not necessarily appreciate that mortality is a consequence of fracture. The current crisis of undertreatment in osteoporosis is thought partly due to the lack of appreciation of mortality risk associated with a fracture.

We think that it is time to change the way fracture risk is communicated. Traditionally, the risk of fracture is conveyed as the probability of sustaining a fracture over a period of time, usually 5 or 10 years. However, given that a fracture is the key risk factor for subsequent fractures and mortality, the *new compound risk* of fracture should be a combination of the risk that someone will sustain a fracture, and the risk that, once they have sustained a fracture, they will sustain further fractures and die.

That thinking guides this study, in which we developed a Markovian model that helps doctors quantify the risk of subsequent fractures and mortality following a fracture for an individual patient.

Please address the reviewers’ comments and questions below.Title: Please see the suggestions of reviewer 3 below for possible revision of the title itself for the paper.

We have modified the short title to read as follows: "Predicting mortality and fracture in patients with osteoporosis." We have also changed the title to read "Epidemiological transition to mortality and refracture following an initial fracture".

Reviewer #1:This is an interesting further analysis of the Dubbo osteoporosis study. Its headline conclusions are easily understood, though the methodology involves esoteric mathematics that will not be readily accessible to clinicians. However, it will be of interest to epidemiologists. My specific questions are as follows:1) The concept of instantaneous risk will be unfamiliar to many readers. It needs to be explained in more detail to make these sections of the manuscript understandable.

As the reviewer notes, the concept of instantaneous risk is not quite accessible to most readers, but it is the basic concept in survival analysis that has been applied in virtually all scientific disciplines. The concept of instantaneous risk was defined in the “Data Analysis” subsection as follows:

“In the multistate model, the transition from state *r* to *s* is governed by *transition intensities* (*q_rs_*) which are estimated from the observed data using the maximum likelihood method. These transition intensities represent the instantaneous risk of moving from current state (state *r*) to the next state (state *s*).”

We have added an explanation in the “Data Analysis” subsection:

"Instantaneous risk (also referred to as “hazard”) is the probability that an individual would experience an event at a particular given point in time. This risk is very small over a very short time period, but can accumulate over time. Thus, the incidence of an event is the product of instantaneous risk and length of time*."*

An example was now added for more details as follows:

“At a given time point, for example, a fracture-free individual (state 1) had an instantaneous risk of staying at that state 1 (*q_11_*), an instantaneous risks of suffering initial fracture (*q_12_*), and an instantaneous risk of death (*q_15_*). These instantaneous risks were estimated for an individual based on the individual’s risk profile and adjusted for age. From these instantaneous risks, we can quantify the individualised risk of fracture, refracture, and mortality for an individual”.

2) Subsection “Instantaneous risk of transition between health states”, where instantaneous risks are being presented, have these calculations been adjusted for age? Those who have already had one, two or three fractures will on average be older than those with fewer fractures and so this will impact on their risk of the next fracture or of death.

The reviewer raises an important point. The instantaneous risk has been adjusted for age. This is now clarified in the manuscript in the “Data Analysis” subsection as follows:

*"*These instantaneous risks were estimated for individuals based on the individual’s risk profile and adjusted for age."

3) The discussions of "typical 70 year olds" are problematic where risk at a T-score of 0 is contrasted with a risk of a T-score of -2.5. T=0 is certainly not typical of a 70 year old, where the average value is closer to -2.0.

We agree with the reviewer's suggestion, and have re-estimated the risk for T-score = -1.5 (Table 2 and Table 4).

Reviewer #2:The manuscript "Predicting mortality and refracture following an initial fracture" by Ho-Le and colleagues reports on data from the Dubbo cohort that examines first fracture, subsequent fracture and mortality in 2046 women and 1205 men followed for 20 years. The authors report that 31% of the women and 15% of the men sustain a first fracture. The risk of the second fracture is 36% in the women and 22% of the men with a mortality that is higher in the men (41%) than in the women (25%). The key predictors of mortality are being male (HR 2.4), old (HR 1.67), and having fem neck BMD (1.16) – all with acceptable confidence intervals. The authors make a prediction model in an effort to provide an evidence-based treatment decision support. The authors feel the impact of their work will be to "change the perception of osteoporosis" because their work is pointing out the "mortality risk".

The link between mortality and osteoporosis is not widely appreciated. In this study we provide evidence from a well characterized cohort to show that fracture does increase the risk of mortality, and that risk should be part of the fracture risk assessment.

We consider that the communication of fracture risk should incorporate mortality risk. Until now, the risk of fracture has been conveyed to patients as the probability of having a fracture over period of time. However, we think that this is inadequate, because the risk should encompass the risks of subsequent fractures and mortality. The new *compound risk* of fracture should therefore be a combination of the risk that someone will sustain a fracture, and the risk that, once they have sustained a fracture, they will sustain further fractures and die. This work contributes to that research direction.

1) Abstract:a) The authors are working on an important area in osteoporosis care – the impact of initial fractures in the elderly and of subsequent fractures on mortality. There is no question mortality is key, but many of the observations that are made and noted in the Abstract of the paper are not new. Male gender and age and BMD – all are known to correlate with poorer outcomes and recurrent fractures in osteoporosis. All the data supporting FRAX and other calculators use these variables. This is not new.

Our focus is not on risk factors for fracture (which is not new); this work focuses on the risk factors for *transition* between health states (e.g. no fracture to fracture, fracture to refracture, fracture to death, etc). Both Garvan Fracture Risk Calculator and FRAX at present estimate the risk of fracture, not the risk of refractures or mortality. Thus, as mentioned above, we consider that our work is unique and will pave way for a new generation of models for fracture risk assessment.

b) Please re-write or revise the sentences in your Abstract as it is not crystal clear what you mean. Is the 31% meaning 31% of the women (634 women)? Did 634 women have a fracture and over what period of time? I think you will need to state the period of observation. Was it the full 20 years? Similarly, with the men (~181 men), when did this 15% have their fractures? Was it 20 years into the collection of data or earlier? Please state this.

We have rewritten the Abstract, including the mentioned sentence: "During the 20-year follow-up period, among 632 women and 184 men those with a first incident fracture, the risk of sustaining a second fracture was higher in women (36%) than men (22%), but mortality risk was higher in men (41%) than women (25%).” This is shown in the paper's new diagram (Figure 1) and the updated Supplementary file 2.

c) When you say the patients sustained a second fracture, what does 36% (women) and 22% (men) refer to? Is this referring back to the original group of women – i.e., 36% of the 634, and for men 22% of the 181 men? Please sharpen up the numbers and clarify the detail. The numbers may be in one of the tables, but it should be clear from the Abstract what you are talking about.

The figures of 36% (in women) and 22% (in men) refer to the incidence of second fracture among 632 women and 184 men with an initial incident fracture. This is now clarified in the revised Abstract and the new diagram (Figure 1).

d) eTable 1 and 2: the numbers are all different from what you are presenting in the Abstract. Your Abstract needs to be understandable on its own, since most readers only read the Abstract (and then cite or not a paper).

We thank the reviewer for this comment. The data mentioned in the Abstract are consistent with eTable 2. However, eTable 2 gives more specific data than the text mentioned. We have rewritten the Abstract, updated the eTable 2 (now mentioned as Supplementary file 2), and added a data flow (Figure 1) to demonstrate the number of each state.

2) When you refer to “eTable 2”, where are the numbers for death following a fragility fracture – 262 women, 105 men. Or are you saying that of the patients who died overall – 627 women, 501 men – these are the %'s of women and men (262 and 105). And the numbers mentioned are "data not shown" in the tables?

The numbers (262 women and 105 men) were derived as the total numbers of women and men who died after an initial fracture + second fracture + subsequent fractures, which were 156+60+46 = 262 for women and 75+22+8 = 105 for men. We have updated eTable 2 (now mentioned as Supplementary file 2) and added a flowchart (Figure 1) to clarify the numbers.

3) eTable 3: what is "instantaneous risk" of transition? Please consider including this definition in the legend to this table. Also please explain what is in the legend. This reviewer does not believe that a general reader will understand (reading the "Note" at the bottom of the table) what was done to generate the numbers in this table. It seems like it will not be clear to the general reader why this set of assumptions were selected. How do the instantaneous risks lead to the Hazard Ratio's in the final column of Table 3 or are they unrelated variables?

We thank the reviewer for this comment. We have moved eTable 3 to the main text (Table 2). We have added an explanation of the concept of instantaneous risk in the “Data Analysis” subsection, which are shown in reviewer 1, comment 1. We have updated the legend of Table 2 to read as follows:

“"Fx", fracture. CI, Confidence interval. […] In each cell, values are percentage of risk and 95% confident interval (in the brackets).”

The hazard ratios in the final column of the table show the ratios of the risks between men and women at each transition, adjusted for age, BMD, history of fall, prior fracture, and comorbidities. This has been clarified in the legend of Table 2.

4) In the presentation of the results in Figure 2A, if the reviewer is reading the data correctly you are seeing an increased risk of transition to fracture/refracture with history of RA for both men and women, which could be highlighted in the paper. The other factors do include age, FN BMD, and prior fracture which you have mentioned in the Abstract and in the subsection “Risk factors for transition between states”. Of interest, neuromuscular disease and COPD are important predictors of fracture/refracture in men and not in women. Was the statistical power (# of events in this modest sized cohort) sufficient to be definitive about the predictive value of the specific clinical risks and disorder (i.e., enough cases, etc)? This question is for Figure 2A and 2B.

The reviewer is correct that women and men with RA had higher risk of transition to fracture/refracture. We have highlighted this finding in the Results section to read as follows:

“In both women and men, having rheumatoid arthritis were more likely to have increased risk of initial fracture and second fracture”.

However, the main purpose of this study was to investigate the impact of advancing age and BMD on the risk of fracture/refracture/mortality. Because all of the concomitant diseases were based on self-report, we were not quite confident in the estimates of association.

5) What does the sentence "The predicted probability of mortality (e.g., fracture, refracture or mortality) for women and men…." Do you mean morbidity and mortality, because when you say "mortality" and then say for example – fracture, refracture or mortality – it is not clear what you mean.

We thank the reviewer for his/her comment. The sentence *"*The predicted probability of mortality.…*"* refers to the adjusted cumulative probability of mortality by fracture status. We have removed the sentence because it is confusing.

6) Figure 3: this is a very important figure in your paper. It is not clear to the reviewer why you chose to do the hypothetical T score of 0 and -2.5 in the text and then chose to do T scores of +0.5 and -2.5 in the Figure 3. Please explain.

The original choice of T-scores was for the purpose of risk contrasting between an individual with normal BMD (T-score = 0) and an osteoporotic individual (T-score = -2.5). However, we think the reviewer 3's point is valid, and we have set the T-score at -1.5 which is the average BMD T-score value for the 70-year old women and men. Thus, Figure 3 has now been modified to show the contrast between T-score of -1.5 and -2.5.

Also the figure legend describes 5 bone health states but there seem to be only 4 states in the boxes (state 1,2,3,4).

Thank you. State 5 is death which is shown in Figure 3. We have corrected the legend of Figure 3 and Figure 3—figure supplement 1 to read as follows:

“There were 4 potential bone heath states an individual could stay before transiting to ***state 5*** (mortality): ***state 1***: No fracture (blue area) if the individual entered the study without any osteoporotic fracture; ***state 2***: initial fracture (light blue area) if an individual had sustained a fracture after study entry; s***tate 3***: Second fracture (orange area) if an individual had suffered a second fracture; and s***tate 4***: Third and further fractures (red area) if an individual had suffered two or more subsequent fractures during the follow-up period.”

In the legend there is no mention of orange, there is mention of green, and there is no green apparent in the figure and there are two different blue's. There needs to be some attention given to clarifying the boxes, the legend and the figure itself and harmonizing it all.

The legend has now been corrected as above. Thank you.

What does "event age" mean? Please define it.

Event age is the age of an individual at the beginning of each state. We have added this definition to the “Data Analysis” subsection to read as follows:

“Age was included in the model as a time-variant factor, and for simplicity, age henceforth is referred to as *age at event*. […] Immortal time bias could occur in groups where individuals transitioned through several states as they had to survive long enough to be able to make these transitions.”

When you label these "age 60" vs. "age 80", do you mean this is the probability for the next 20 year for mortality (i.e., age 60-80, and ages 80-100)?

That “age 60” or “age 80” is the age of an individual at the beginning of each heath state. We estimated the risk that an individual may die at any time within t years from the current state, t ranges from 0 to 20 as shown in the X axis of Figure 3. For example, t = 5 means 5-year probability of mortality for women and men at the age of 60 (i.e. from age 60 to 65) or for women and men at the age of 80 (i.e. from age 80 to 85).

If all subjects had to be 60 years old + to get into the study, there would be relatively few who were age 60 when their event (fracture) occurred from which to generate these curves. Could you provide insight into what data you had that went into the “age 60” as well as age 80? It is appreciated that these are estimates but what could you be estimating from at age 60 when you are talking about State 3 or 4? It is expected that it would take some years to get that 2nd, 3rd or 4th fracture.

The reviewer is correct that it takes some years to move from one state to another (eg fracture to mortality). As explained above, we used the “event age” at the begining of each health state as an input variable. We have provided in Author response table 1 data on fracture incidence stratified by age groups for your perusal.

**Author response table 1. resptable1:** 

	Event age (years)				
	60-69	70-79	≥80	Total	
Baseline age (years)	60-69	158	209	68	435
	70-79	-	115	171	286
	≥80	-	-	95	95
	Total	158	324	334	816

7) Discussion:a) The progression to premature mortality after a fracture could be due to the acquisition over time of other co-morbid conditions that occur in aging. Are you sure that strokes, infarct, cancer, infections/sepsis/COVID-19 etc did not occur in the years post a fracture? After reading the Materials and methods it is not clear that once a subject is enrolled that he or she would be removed from analysis if terminal cancer developed.

We thank the reviewer for his/her comment. We only adjusted for baseline comorbidities, because we have not ascertained the incidence of comorbidities over time. We have discussed this as a potientalweakness of study:

“Comorbidities occurred during the follow-up time were not ascertained, and could not treat as time-variant covariates in the analysis.”

Granted you have presumably the same rates of cancer across the board, for example, but some of these groups may not be so large – 2, 3, 4 fractures. There would be no reason to anticipate that terminal events tracked with fractures especially if the fracture was years before or tracked with the fractures. There has been some attention to this issue with hip fractures because of the sarcopenia and disability that they cause – perhaps mortality is related to that attendant frailty. However, here you are counting all fragility fractures and it is hard to see that they would have the same impact as hip fracture on the overall health.

We agree that the increased mortality risk after a hip fracture could be attributed to frailty and comorbidities. However, a number of previous studies have shown that even after accounting for frailty and comorbidities, patients with a hip fracture still have excess mortality risk.

b) In the sixth paragraph of the Discussion you are discussing estimating fracture risk and mortality risk, but you then say that patients "usually underestimate their overall health". Do you mean overestimate their health or underestimate their risk of death?

The reviewer is correct. We have rewritten the text under the Discussion section as follows:

"Patients do not always appreciate the serious consequence of fracture (e.g. subsequent fractures and mortality), and as a result, they usually underestimate their risk of adverse outcomes".

c) While you have provided models of risk of death, you have not established causality between the fractures and mortality to my review of the paper. Therefore it seems an over-strong statement to say – "…providing the estimated risk of mortality can motivate them to take preventive measures." In an non-interventional, observational study, you cannot assume that fracture prevention will significantly change the mortality risks, curves.

Although we could monitor the movement between health states, we could not make any causal inference from the data, because this is an observational study. However, there are several lines of evidence (from randomized controlled trials [RCT]) that treating patients with osteoporosis and/or fracture reduces their risk of fracture by approximately 50%. We also have RCT evidence that treating patients with a hip fracture reduces their risk of mortality by between 28 and 35%. Thus, these evidence into context, we consider that our statement is not that strong. We have mentioned these studies in the Discussion section as follows:

“Evidence from randomized controlled trials suggest that in patients with osteoporosis and/or a pre-existing fracture, bisphosphonate treatment reduces fracture risk and mortality risk. […] However, a recent meta-analysis found that the effect of bisphosphonate treatment on mortality was less certain (Cummings et al., 2019).”

d) Were patients in this cohort ever given any treatments and how did you handle concomitant medications such as estrogen, bisphosphonates, SERM, testosterone and so forth? If the cohort was started in 1989 (21 years would be through 2009), and repeated fractures were occurring in hundreds of these patients, were they treated, if so with what and how was that handled in the data analysis? Were individuals specifically not treated, which would be an ethical issue? You also state (subsection “Participants and Setting”) that the study is ongoing. Surely by now subjects in the cohort have been receiving treatments for osteoporosis.

A small proportion (<2% of participants) has been on any anti-osteoporosis treatment. Due to the small sample size of treated individuals, we did not consider treatment in the model of analysis.

e) It would be of interest to the reader to know the breakdown of fractures that you counted. How many were clinical vertebral, hip, humerus, rib, tibia, fibula, wrist etc. What kinds of fractures are these models resting on? What are the causes of death in the cohort?

The incidence of fractures stratified by gender is shown in Author response table 2:

**Author response table 2. resptable2:** 

Fracture type	Women	Men	Total
Hip fracture	87	29	116
Vertebral fracture	240	80	320
Non-hip non-vertebral fracture	305	75	380
Total	632	184	816

We could not ascertain the cause of death in the study, and this is a weakness that we have mentioned in the Discussion section to read as follows:

“However, the cause of death was not available, and it was not possible to conduct an in-depth analysis of mortality attributable risk.”

8) The reports were reviewed, but were the fractures adjudicated? Were all deaths confirmed? (Subsection “Fracture ascertainment”).

Yes, all deaths were confirmed and all fractures were adjudicated. These information have been clarified in the subsection “Fracture ascertainment”, as follows:

"Fractures occurring during the study period were identified through radiologists’ reports from the only two radiology centers providing X-ray services within the Dubbo region as previously described. […] In this study, we included only fractures having definite reports and resulting from low-energy trauma such as falls from standing height or less in this analysis."

“The incidence of death was ascertained by the NSW Registry of Births, Death and Marriages. Deaths were also monitored by systematically searching funeral director lists, local newspapers and on radio or by word of mouth with a confirmation or bi-annual telephone contact.”

9) One wonders if Figure 1 is needed and whether the figures mainly containing the dots are needed. Please consider another shorter way to convey this information.

Thank you for the suggestion. The dot figures visualize the data shown in Table 2. However, we agree with the reviewer that Figure 1 is no longer required.

10) Please read the legend to eTable 4. Please revise "the person who ages of 70". There is a misspelling in the legend to Supplementary eFigure 1.

Thank you. We have moved the eTable 4 to the main text (Table 3) and the legend is now rewritten as follows: "at the age of 70". The eFigure 1 has been moved to the main text (Figure 4) and the legend has now been corrected to read as follows:

**“**Figure 4: Markovian model of transition between 5 heath states (e.g. alive without a fracture, initial fracture, second fracture, third and further fracture, and death). […] At a given time point, for example, a fracture-free individual (state 1) had an instantaneous risk of staying at that state 1 (*q_11_*), an instantaneous risks of suffering initial fracture (*q_12_*), and an instantaneous risk of death (*q_15_*)”.

Reviewer #3:The authors used data from the Dubbo Osteoporosis Epidemiology Study to analyze risk for initial fracture, subsequent fractures and mortality in older men and women followed for 20 years. They found that fractures were common, occurred earlier in women than in men, and were associated with earlier mortality. Mortality risk following a fracture was higher in men compared to women. The authors presented a detailed analysis of refracture and mortality rates in men and women following fracture at age 60 and age 80, to create a prediction model.

Thank you for taking time to consider our manuscript. We think that mortality should be part of the doctor – patient communication. The "risk of fracture" should encompass the probability of sustaining a future fracture, and one a fracture has occurred, the probability of mortality. That thinking underlies the model developed here.

1) The major findings of this study are not new. We already know that fractures occur at an earlier age in women compared to men, and that mortality following fracture is higher in men than women. We also know that advancing age and lower BMD are associated with higher risk for fracture. This study presents a more detailed analysis of risk for subsequent fracture and mortality following a fracture than previous studies have done.

We thank the reviewer for his/her comment. The focus of this study is not on the association between fracture and mortality which we and others have previously shown. Our focus is on the development of a new prediction model for quantifying the risks of *transition* between health states (e.g. fracture to refracture, fracture to death, refracture to death, etc). This Markovian model is first introduced in the field of osteoporosis research. In the Discussion, we also introduce the idea of “skeletal age” for fracture and mortality risk communication.

2) The author claim to have developed a prediction model to help patients and doctors make evidence-based treatment decisions. I did not find this study to offer a clinically useful risk prediction model, nor does this study address osteoporosis treatment. It does offer new information about mortality following fracture. Life expectancy is not commonly discussed in osteoporosis care, in the way it may be discussed in cancer treatment, for example. Given the average age of participants in this study, and without presenting evidence that osteoporosis treatment can influence mortality rates, I am not sure that these data will be readily incorporated into osteoporosis treatment discussions.

The prediction model is shown in Table 2, Table 4, Figures 2A and B, and illustrated by Figure 4. In the revised manuscript, we further provide an example of interpretation in terms of what we call “skeletal age” (Discussion section). For example, for a 70 years old man who has sustained a fracture, the hazard ratio of 1.67 is equivalent to a loss of 5.4 years of life. In other words, for the 70 years old man, the hazard ratio of 1.67 corresponds to a skeletal age of 75.4 years.

Several lines of evidence suggest that bisphosphonate (the first line pharmacologic treatment for osteoporosis) is associated with reduced mortality risk. For instance, data from randomized controlled clinical trial show that in osteoporotic patients with a preexisting fracture, zoledronic acid reduces fracture risk and mortality risk (Lyles et al., 2007; Reid et al., 2018). The beneficial effect of treatment (relative risk reduction of mortality) can also be expressed in terms of “skeletal age”. We have discussed this point in the manuscript (Discussion section) as follows:

“Evidence from randomized controlled trials suggest that in patients with osteoporosis and/or a pre-existing fracture, bisphosphonate treatment reduces fracture risk and mortality risk. […] Taken together, these results suggest that in osteoporotic or osteopenic patients with a fracture, bisphosphonate treatment may reduce mortality risk. The beneficial effect of treatment can also be expressed in terms of “skeletal age”.”

We also revised the conclusion of the Abstract to read as follows:

"These results were incorporated into a prediction model to aid patient-doctor discussion about fracture vulnerability and treatment decisions."

In the future, we will also implement a web based risk calculator that incorporates life expectancy into fracture risk assessment.

3) Impact Statement: "Our results will change the perception of osteoporosis because it increases mortality risk. The personalized risk prediction model presented here will change the assessment of fracture risk in the future."Many patients are diagnosed with osteoporosis based on T-score criteria (no prior fracture). Figure 2B shows that in those without prior fracture, BMD is not associated with mortality. Perhaps this impact statement should read "osteoporotic fracture increases mortality risk".

We have modified the Impact Statement as follows:

"The concept of compound fracture risk is redefined to combine the risk that an individual will sustain a fracture, and the risk of mortality once a fracture has occurred."